# CLN2INV: Learning Loop Invariants with Continuous Logic Networks

**Gabriel Ryan**[†*]**, Justin Wong**[†*]**, Jianan Yao**[†*]**, Ronghui Gu**[†‡]**, Suman Jana**[†]

[†]Department of Computer Science, Columbia University, New York, NY, USA
{gabe,jianan,rgu,suman}@cs.columbia.edu
{justin.wong}@columbia.edu
[‡]CertiK, New York, NY, USA
{ronghui.gu}@certik.org

## Abstract

Program verification offers a framework for ensuring program correctness and therefore systematically eliminating different classes of bugs. Inferring loop invariants is one of the main challenges behind automated verification of real-world programs, which often contain many loops. In this paper, we present the Continuous Logic Network (CLN), a novel neural architecture for automatically learning loop invariants directly from program execution traces. Unlike existing neural networks, CLNs can learn precise and explicit representations of formulas in Satisfiability Modulo Theories (SMT) for loop invariants from program execution traces. We develop a new sound and complete semantic mapping for assigning SMT formulas to continuous truth values that allows CLNs to be trained efficiently. We use CLNs to implement a new inference system for loop invariants, CLN2INV, that significantly outperforms existing approaches on the popular Code2Inv dataset. CLN2INV is the first tool to solve all 124 theoretically solvable problems in the Code2Inv dataset. Moreover, CLN2INV takes only 1.1 second on average for each problem, which is 40× faster than existing approaches. We further demonstrate that CLN2INV can even learn 12 significantly more complex loop invariants than the ones required for the Code2Inv dataset.

## 1 Introduction

Program verification offers a principled approach for systematically eliminating different classes of bugs and proving the correctness of programs. However, as programs have become increasingly complex, real-world program verification often requires prohibitively expensive manual effort (Wilcox et al., 2015; Gu et al., 2016; Chajed et al., 2019). Recent efforts have focused on automating the program verification process, but automated verification of general programs with unbounded loops remains an open problem (Nelson et al., 2017; 2019).

Verifying programs with loops requires determining *loop invariants*, which captures the effect of the loop on the program state irrespective of the actual number of loop iterations. Automatically inferring correct loop invariants is a challenging problem that is undecidable in general and difficult to solve in practice (Blass & Gurevich, 2001; Furia et al., 2014). Existing approaches use stochastic search (Sharma & Aiken, 2016), heurstics-based search (Galeotti et al., 2015), PAC learning based on counterexamples (Padhi & Millstein, 2017), or reinforcement learning (Si et al., 2018). However, these approaches often struggle to learn complex, real-world loop invariants.

In this paper, we introduce a novel approach to learning loop invariants by modeling the loop behavior from program execution traces with a new type of neural architecture. We note that inferring loop invariants can be posed as learning formulas in Satisfiability Modulo Theories (SMT) (Biere et al., 2009) over program variables collected from program execution traces (Nguyen et al., 2017). In principle, neural networks are well suited to this task because they can act as universal function

---

[*]Co-student leads listed in alphabetical order; each contributed equally.

approximators and have been successfully applied in various domains that require modeling of arbitrary functions (Hornik et al., 1989; Goodfellow et al., 2016). However, loop invariants must be represented as explicit SMT formulas to be usable for program verification. Unfortunately, existing methods for extracting logical rules from general neural architectures lack sufficient precision (Augasta & Kathirvalavakumar, 2012), while inductive logic learning lacks sufficient expressiveness for use in verification (Evans & Grefenstette, 2018).

We address this issue by developing a novel neural architecture, Continuous Logic Network (CLN), which is able to efficiently learn explicit and precise representations of SMT formulas by using continuous truth values. Unlike existing neural architectures, CLNs can represent a learned SMT formula explicitly in its structure and thus allow us to precisely extract the exact formula from a trained model.

In order to train CLNs, we introduce a new semantic mapping for SMT formulas to continuous truth values. Our semantic mapping builds on BL, or basic fuzzy logic (Hájek, 2013), to support general SMT formulas in a continuous logic setting. We further prove that our semantic model is sound (i.e., truth assignments for the formulas are consistent with their discrete counterparts) and complete (i.e., all formulas can be represented) with regard to the discrete SMT formula space. These properties allow CLNs to represent any quantifier-free SMT formula operating on mixed integer-real arithmetic as an end-to-end differentiable series of operations.

We use CLNs to implement a new inference system for loop invariants, CLN2INV, that significantly outperforms state-of-the-art tools on the Code2Inv dataset by solving all 124 theoretically solvable problems in the dataset. This is 20 problems more than LoopInvGen, the winner of the SyGus 2018 competition loop invariant track (Padhi & Millstein, 2017). Moreover, CLN2INV finds invariants for each program in 1.1 second on average, more than 40 times faster than LoopInvGen. We also demonstrate that CLN2INV is able to learn complex, real-world loop invariants with combinations of conjunctions and disjunctions of multivariable constraints. Our source code and benchmarks are publicly available on Github[1].

Our main contributions are:

- We introduce a new semantic mapping for assigning continuous truth values to SMT formulas that is theoretically grounded and enables learning formulas through backpropagation. We further prove that our semantic model is sound and complete.

- We develop a novel neural architecture, Continuous Logic Networks (CLNs), that to the best of our knowledge is the first to efficiently learn precise and explicit SMT formulas by construction.

- We use CLNs to implement a new loop invariant inference system, CLN2INV, that is the first to solve all 124 theoretically solvable problems in the Code2Inv dataset, 20 more than the existing methods. CLN2INV is able to find invariants for each problem in 1.1 second on average, $40\times$ faster than existing systems.

- We further show that CLN2INV is able to learn 12 more complex loop invariants than the ones present in the Code2Inv dataset with combinations of multivariable constraints.

**Related Work.** Traditionally, loop invariant learning relies on stochastic or heuristics-guided search (Sharma & Aiken, 2016; Galeotti et al., 2015). Other approaches like NumInv analyze traces and discover conjunctions of equalities by solving a system of linear equations (Sharma et al., 2013; Nguyen et al., 2017). LoopInvGen uses PAC learning of CNF using counterexamples (Padhi et al., 2016; Padhi & Millstein, 2017). By contrast, Code2Inv learns to guess loop invariants using reinforcement learning with recurrent and graph neural networks (Si et al., 2018). However, these approaches struggle to learn complex invariants. Unlike these works, CLN2INV efficiently learns complex invariants directly from execution traces.

There is extensive work on PAC learning of boolean formulas, but learning precise formulas requires a prohibitively large number of samples (Kearns et al., 1994). Several recent works use differentiable logic to learn boolean logic formulas from noisy data (Kimmig et al., 2012; Evans & Grefenstette, 2018; Payani & Fekri, 2019) or improving adversarial robustness by applying logical rules to training

---

[1]https://github.com/gryan11/cln2inv

(Fischer et al., 2019). By contrast, our work learns precise SMT formulas directly by construction, allowing us to learn richer predicates with compact representation in a noiseless setting.

A variety of numerical relaxations have been applied to SAT and SMT solving. Application-specific approximations using methods such as interval overapproximation and slack variables have been developed for different classes of SMT (Eggers et al., 2008; Nuzzo et al., 2010). More recent work has applied recurrent and graph neural networks to Circuit SAT problems and unsat core detection (Amizadeh et al., 2019; Selsam et al., 2019; Selsam & Bjørner, 2019). FastSMT uses embeddings from natural language processing like skip-gram and bag-of-words to represent formulas for search strategy optimization (Balunovic et al., 2018). Unlike these approaches, we relax the SMT semantics directly to generate a differentiable representation of SMT.

## 2 BACKGROUND

In this section, we introduce the problem of inferring loop invariants and provide a brief overview of Satisfiability Modulo Theories (SMT), which are used to represent loop invariants. We provide background into fuzzy logic, which we extend with our new continuous semantic mapping for SMT.

### 2.1 LOOP INVARIANTS

Loop invariants capture loop behavior irrespective of number of iterations, which is crucial for verifying programs with loops. Given a loop, $\text{while}\,(LC)\,\{C\}$, a precondition $P$, and a post-condition $Q$, the verification task involves finding a loop invariant $I$ that can be concluded from the precondition and implies the post-condition (Hoare, 1969). Formally, it must satisfy the following three conditions, in which the second is a Hoare triple describing the loop:

$$P \implies I \qquad \{I \wedge LC\}\, C\, \{I\} \qquad \neg LC \wedge I \implies Q$$

**Example of Loop Invariant.** Consider the example loop in Fig.1. For a loop invariant to be usable, it must be valid for the precondition $(t = 10 \wedge u = 0)$, the inductive step when $t \neq 0$, and the post-condition $(u = 20)$ when the loop condition is no longer satisfied, i.e., $t = 0$. The correct and precise invariant $I$ for the program is $(2t + u = 20)$.

```
//pre: t=10 /\ u=0
while (t != 0){
  t = t - 1;
  u = u + 2;
}
//post: u=20
```

The desired loop invariant $I$ for the left program is a boolean function over program variables $t, u$ such that:

$$\forall t\, u, \begin{cases} t = 10 \wedge u = 0 & \implies I(t, u) & (pre) \\ I(t, u) \wedge (t \neq 0) & \implies I(t - 1, u + 2) & (inv) \\ I(t, u) \wedge (t = 0) & \implies u = 20 & (post) \end{cases}$$

(a) Example loop

(b) The desired and precise loop invariant $I$ is $(2t + u = 20)$.

Figure 1: Example Loop Invariant inference problem.

### 2.2 SATISFIABILITY MODULO THEORIES

Satisfiability Modulo Theories (SMT) are an extension of Boolean Satisfiability that allow solvers to reason about complex problems efficiently. Loop invariants and other formulas in program verification are usually encoded with quantifier-free SMT. A formula $F$ in quantifier-free SMT can be inductively defined as below:

$$F := E_1 \bowtie E_2 \mid \neg F \mid F_1 \wedge F_2 \mid F_1 \vee F_2 \qquad \bowtie\, \in \{=, \neq, <, >, \leq, \geq\}$$

where $E_1$ and $E_2$ are expressions of terms. The loop invariant $(2t + u = 20)$ in Fig. 1 is an SMT formula. Nonlinear arithmetic theories admit higher-order terms such as $t^2$ and $t * u$, allowing them to express more complex constraints. For example, $(\neg(2 \geq t^2))$ is an SMT formula that is true when the value of the high-order term $t^2$ is larger than 2.

## 2.3 BASIC FUZZY LOGIC (BL)

Basic fuzzy logic (BL) is a class of logic that uses continuous truth values in the range $[0, 1]$ and is differentiable almost everywhere[2] (Hájek, 2013). BL defines logical conjunction with functions called *t-norms*, which must satisfy specific conditions to ensure that the behavior of the logic is consistent with boolean First Order Logic. Formally, a t-norm (denoted $\otimes$) in BL is a binary operator over truth values in the interval $[0, 1]$ satisfying the following conditions:

1. *associativity* and *commutativity*: the order in which a set of t-norms on continuous truth values are evaluated should not change the result: $x \otimes (y \otimes z) = (x \otimes y) \otimes z$ and $x \otimes y = y \otimes x$.

2. *monotonicity*: increasing any input value to a t-norm operation should not cause the result to decrease: $x_1 \leq x_2 \implies x_1 \otimes y \leq x_2 \otimes y$

3. *consistency*: the result of any t-norm applied to a truth value and 1 should be 1, and the result of any truth value and 0 should be 0: $1 \otimes x = x$ and $0 \otimes x = 0$

Besides these conditions, BL also requires that t-norms be continuous. Given a t-norm $\otimes$, its associated *t-conorm* (denoted $\oplus$) is defined with DeMorgan's law: $t \oplus u \triangleq \neg(\neg t \otimes \neg u)$, which can be considered as logical disjunction. A common t-norm is the product t-norm $x \otimes y = x \cdot y$ with its associated t-conorm $x \oplus y = x + y - x \cdot y$.

## 3 CONTINUOUS SATISFIABILITY MODULO THEORIES

We introduce a *continuous semantic mapping*, $\mathcal{S}$, for SMT on BL that is end-to-end differentiable. The mapping $\mathcal{S}$ associates SMT formulas with continuous truth values while preserving each formula's semantics. In this paper, we only consider quantifier-free formulas. This process is analogous to constructing t-norms for BL, where a t-norm operates on continuous logical inputs.

We define three desirable properties for continuous semantic mapping $\mathcal{S}$ that will preserve formula semantics while facilitating parameter training with gradient descent:

1. $\mathcal{S}(F)$ should be consistent with BL. For any two formulas $F$ and $F'$, where $F(x)$ is satisfied and $F'(x)$ is unsatisfied with an assignment $x$ of formula terms, we should have $\mathcal{S}(F')(x) < \mathcal{S}(F)(x)$. This will ensure the semantics of SMT formulas are preserved.

2. $\mathcal{S}(F)$ should be differentiable almost everywhere. This will facilitate training with gradient descent through backpropogation.

3. $\mathcal{S}(F)$ should be increasing everywhere as the terms in the formula approach constraint satisfaction, and decreasing everywhere as the terms in the formula approach constraint violation. This ensures there is always a nonzero gradient for training.

**Continuous semantic mapping.** We first define the mapping for ">" (greater-than) and "$\geq$" (greater-than-or-equal-to) as well as adopting definitions for "$\neg$", "$\wedge$", and "$\vee$" from BL. All other operators can be derived from these. For example, "$\leq$" (less-than-or-equal-to) is derived using "$\geq$" and "$\neg$", while "$=$" (equality) is then defined as the conjunction of formulas using "$\leq$" and "$\geq$." Given constants $B > 0$ and $\epsilon > 0$, we first define the the mapping $\mathcal{S}$ on ">" and "$\geq$" using shifted and scaled sigmoid functions:

$$\mathcal{S}(t > u) \triangleq \frac{1}{1 + e^{-B(t-u-\epsilon)}} \qquad \mathcal{S}(t \geq u) \triangleq \frac{1}{1 + e^{-B(t-u+\epsilon)}}$$

We illustrate these functions in Figure 2. The validity of our semantic mapping lie in the following facts, which can be proven with basic algebra.

$$\lim_{\substack{\epsilon \to 0^+ \\ B \cdot \epsilon \to +\infty}} \frac{1}{1 + e^{-B(t-u-\epsilon)}} = \begin{cases} 1 & t > u \\ 0 & t \leq u \end{cases} \qquad \lim_{\substack{\epsilon \to 0^+ \\ B \cdot \epsilon \to +\infty}} \frac{1}{1 + e^{-B(t-u+\epsilon)}} = \begin{cases} 1 & t \geq u \\ 0 & t < u \end{cases}$$

---

[2]Almost everywhere indicates the function is differentiable everywhere except for a set of measure 0. For example, a Rectified Linear Unit is differentiable almost everywhere except at zero.

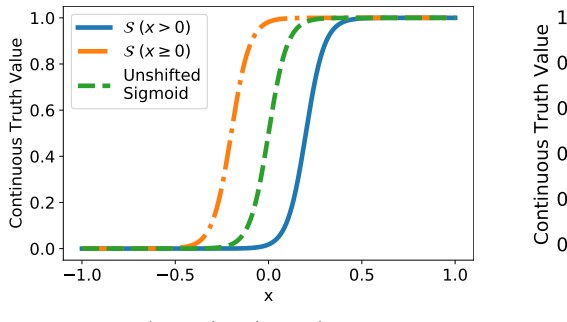 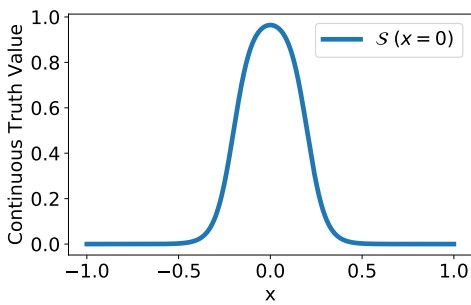

(a) Plot of $\mathcal{S}(x \geq 0)$, $\mathcal{S}(x > 0)$ with sigmoid

(b) Plot of $\mathcal{S}(x = 0)$ with product t-norm

Figure 2: Illustration of the mapping $\mathcal{S}$ on $>, \geq, =$ when $B = 20$ and $\epsilon = 0.2$

When $\epsilon$ goes to zero and $B * \epsilon$ goes to infinity, our continuous mapping of ">" and "$\geq$" will preserve their original semantics. Under these conditions, our mapping satisfies all three desirable properties. In practice, for small $\epsilon$ and large $B$, the properties are also satisfied if $|t - u| > \epsilon$.

Next we define the mapping $\mathcal{S}$ for boolean operators "$\wedge$", "$\vee$" and "$\neg$" using BL. Given a specific t-norm $\otimes$ and its corresponding t-conorm $\oplus$, it is straightforward to define mappings of "$\wedge$", "$\vee$" and "$\neg$":

$$\mathcal{S}(F_1 \wedge F_2) \triangleq \mathcal{S}(F_1) \otimes \mathcal{S}(F_2) \qquad \mathcal{S}(F_1 \vee F_2) \triangleq \mathcal{S}(F_1) \oplus \mathcal{S}(F_2) \qquad \mathcal{S}(\neg F) \triangleq 1 - \mathcal{S}(F)$$

Based on the above definitions, the mapping for other operators can be derived as follows:

$$\mathcal{S}(t < u) = \mathcal{S}(\neg(t \geq u)) = \frac{1}{1 + e^{B(t-u+\epsilon)}} \qquad \mathcal{S}(t \leq u) = \mathcal{S}(\neg(t > u)) = \frac{1}{1 + e^{B(t-u-\epsilon)}}$$

$$\mathcal{S}(t = u) = \mathcal{S}((t \geq u) \wedge (t \leq u)) = \frac{1}{1 + e^{-B(t-u+\epsilon)}} \otimes \frac{1}{1 + e^{B(t-u-\epsilon)}}$$

Figure 2The mapping $\mathcal{S}$ on "=" is valid since the following limit holds (see Appendix A for the proof).

$$\lim_{\substack{\epsilon \to 0^+ \\ B \cdot \epsilon \to +\infty}} \mathcal{S}(t = u) = \lim_{\substack{\epsilon \to 0^+ \\ B \cdot \epsilon \to +\infty}} \frac{1}{1 + e^{-B(t-u+\epsilon)}} \otimes \frac{1}{1 + e^{B(t-u-\epsilon)}} = \begin{cases} 1 & t = u \\ 0 & t \neq u \end{cases}$$

The mapping for other operators shares similar behavior in the limit, and also fulfill our desired properties under the same conditions.

Using our semantic mapping $\mathcal{S}$, most of the standard operations of integer and real arithmetic, including addition, subtraction, multiplication, division, and exponentiation, can be used normally and mapped to continuous truth values while keeping the entire formula differentiable. Moreover, any expression in SMT that has an integer or real-valued result can be mapped to continuous logical values via these formulas, although end-to-end differentiability may not be maintained in cases where specific operations are nondifferentiable.

## 4 Continuous Logic Networks

In this section, we describe the construction of Continuous Logic Networks (CLNs) based on our continuous semantic mapping for SMT on BL.

**CLN Construction.** CLNs use our semantic mapping to provide a general neural architecture for learning SMT formulas. In a CLN, the learnable coefficients and smoothing parameters correspond to the learnable parameters in a standard feedforward network, and the continuous predicates, t-norms, and t-conorms operate as activation functions like ReLUs in a standard network. In this

paper, we focus on shallow networks to address the loop invariant inference problem, but we envision deeper general purpose CLNs that can learn arbitrary SMT formulas. When constructing a CLN, we work from an *SMT Formula Template*, in which every value is marked as either an input term, a constant, or a learnable parameter. Given an SMT Formula Template, we dynamically construct a CLN as a computational graph. Figure 3 shows a simple formula template and the constructed CLN. We denote the CLN model constructed from the formula template $\mathcal{S}(F)$ as $M_F$.

**CLN Training.** Once the CLN has been constructed based on a formula template, it is trained with the following optimization. Given a CLN model $M$ constructed from an SMT template with learnable parameters $\mathbf{W}$, and a set $\mathbf{X}$ of valid assignments for the terms in the SMT template, the expected value of the CLN is maximized by minimizing a loss function $\mathcal{L}$ that penalizes model outputs that are less than one. A minimum scaling factor $\beta$ is selected, and a hinge loss is applied to the scaling factors ($B$) to force the differentiable predicates to approach sharp cutoffs. The offset $\epsilon$ is also regularized to ensure precision. The overall optimization is formulated as:

$$\max_{\{\mathbf{W},B,\epsilon\}} \mathbb{E}[M(\mathbf{X};\mathbf{W},B,\epsilon)] = \min_{\{\mathbf{W},B,\epsilon\}} \sum_{\boldsymbol{x}\in\mathbf{X}} \mathcal{L}(M(\boldsymbol{x};\mathbf{W},B,\epsilon)) + \lambda \sum_{B\in\mathbf{B}} \mathcal{L}_{hinge}(\beta,B) + \gamma||\epsilon||_2$$

where $\lambda$ and $\gamma$ are hyperparameters respectively governing the weight assigned to the scaling factor and offset regularization. $\mathcal{L}_{hinge}(\beta, B)$ is defined as $max(0, \beta - B)$, and $\mathcal{L}$ is any loss function strictly decreasing in domain $[0, 1]$.

Given a CLN that has been trained to a loss approaching 0 on a given set of valid assignments, we show that the resulting continuous SMT formula learned by the CLN is consistent with an equivalent discrete SMT formula. In particular, we prove that such a formula is *sound*, (i.e., a CLN will learn a correct SMT formula with respect to the training data), and that our continuous mapping is *complete*, (i.e., CLNs can represent any SMT formula that can be represented in discrete logic), where these properties are defined for CLNs as follows:

**Soundness.** Given the SMT formula $F$, the CLN model $M_F$ constructed from $\mathcal{S}(F)$ always preserves the truth value of $F$. It indicates that given a valid assignment to the terms $\boldsymbol{x}$ in $F$, $F(\boldsymbol{x}) = True \iff M_F(\boldsymbol{x}) = 1$ and $F(\boldsymbol{x}) = False \iff M_F(\boldsymbol{x}) = 0$.

**Completeness.** For any SMT formula $F$, a CLN model $M$ can be constructed representing that formula. In other words, CLNs can express all SMT formulas on integers and reals.

We further prove that CLNs are guaranteed to converge to a globally optimal solution for formulas, which can be expressed as the conjunction of linear equalities. We provide formal definitions and proofs for soundness and completeness in Appendix B and optimality in Appendix C.

## 5 Loop Invariant Learning

We use CLNs to implement a new inference system for loop invariants, CLN2INV, which learns invariants directly from execution traces. CLN2INV follows the same overall process as other loop invariant inference systems such as LoopInvGen and Code2Inv – it iterates through likely candidate invariants and checks its validity with an SMT solver. The key difference between our method and other systems is that it learns a loop invariant formula directly from trace data. Figure 3 provides an overview of the architecture.

**Preprocessing.** We first perform static analysis and instrument the program to prepare for training data generation. In addition to collecting the given precondition and post-condition, the static analysis extracts all constants in the program, along with the loop termination condition. We then instrument the program to record all program variables before each loop execution and after the loop termination. We also restrict the loop to terminate after a set number of iterations to prevent loops running indefinitely (for experiments in this paper, we set the max loop iterations to 50). We also strengthen the precondition to ensure loop execution (see Appendix D).

**Training Data Generation.** We generate training data by running the program repeatedly on a set of randomly initialized inputs that satisfy the preconditions. Unconstrained variables are initialized from a uniform distribution centered on 0 with width $r$, where $r$ is a hyperparameter of the sampling process. Variables with either upper or lower bound precondition constraints are initialized from a uniform distribution adjacent to their constraints with width $r$, while variables with both upper

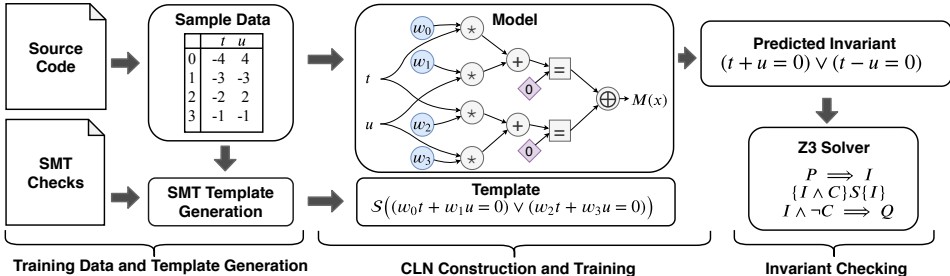

Figure 3: System architecture and CLN construction from SMT templates.

and lower bounds in the precondition are sampled uniformly within their bounds. For all of our experiments in this paper, we set $r$ to 10. When the number of uninitialized variables is small (i.e., less than 3), we perform this sampling exhaustively. An example of training data generation is provided in Appendix E.

**Template Generation.** We generate templates in three stages with increasing expressiveness:

1. We first generate templates directly from the pre-condition and post-condition.

2. We next extract the individual clauses from the pre- and post-condition as well as the loop condition, and generate templates from conjunctions and disjunctions of each possible pair of clauses.

3. We finally generate more generic templates of increasing complexity with a combination of one or more equality constraints on all variables combined with conjunctions of inequality constraints, which are based on the loop condition and individual variables.

We describe the template generation in detail in Appendix E. To detect when higher order terms may be present in the invariant, we perform a log-log linear regression on each variable relative to the loop iteration, similarly to Sharma et al. (2013). If the loop contains one or more variables that grow superlinearly relative to the loop iteration, we add higher order polynomial terms to the equality constraints in the template, up to the highest degree detected among the loop variables.

**CLN Construction and Training.** Once a template formula has been generated, a CLN is constructed from the template using the formulation in §4. As an optimization, we represent equality constraints as Gaussian-like functions that retain a global maximum when the constraint is satisfied as discussed in Appendix F. We then train the model using the collected execution traces.

**Invariant Checking.** Invariant checking is performed using SMT solvers such as Z3 (De Moura & Bjørner, 2008). After the CLN for a formula template has been trained, the SMT formula for the loop invariant is recovered by normalizing the learned parameters. The invariant is checked against the pre, post, and inductive conditions as described in §2.1. If the correct invariant is not found, we return to the template generation phase to continue the search with a more expressive template.

## 6 EXPERIMENTS

We compare the performance of CLN2INV with two existing methods and demonstrate the efficacy of the method on several more difficult problems. Finally, we conduct two ablation studies to justify our design choices.

**Test Environment.** All experiments are performed on an Ubuntu 18.04 server with an Intel Xeon E5-2623 v4 2.60GHz CPU, 256Gb of memory, and an Nvidia GTX 1080Ti GPU.

**System Configuration.** We implement CLNs in PyTorch and use the Adam optimizer for training with learning rate 0.01 (Paszke et al., 2017; Kingma & Ba, 2014). Because the performance of CLN is dependent on weight initialization, the CLN training randomly restart if the model does not reach termination within 2,000 epochs. Learnable parameters are initialized from a uniform distribution in the range [-1, 1], which we found works well in practice.

**Test Dataset.** We use the same benchmark used in the evaluation of Code2Inv. We have removed nine invalid programs from Code2Inv's benchmark and test on the remaining 124. The removed programs are invalid because there are inputs which satisfy the precondition but result in a violation of the post-condition. See Appendix G for details on the removed problems. The benchmark consists of loops expressed as C code and corresponding SMT files. Each loop can have nested if-then-else blocks (without nested loops). Programs in the benchmark may also have uninterpreted functions with boolean return values (emulating external function calls) in branches or loop termination conditions.

## 6.1 COMPARISON TO EXISTING SOLVERS

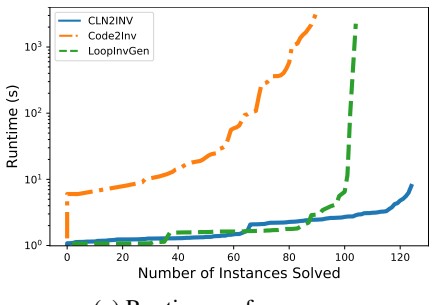 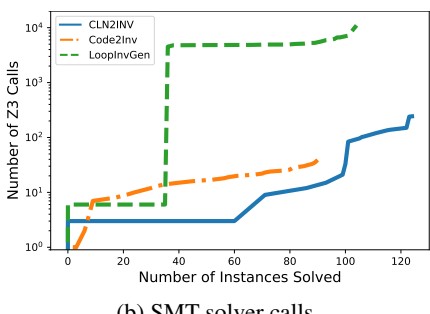

(a) Runtime performance.                     (b) SMT solver calls.

Figure 4: Performance evaluation.

**Performance Comparison.** We compare CLN2INV to two state-of-the-art methods: Code2Inv (based on neural code representation and reinforcement learning) and LoopInvGen (PAC learning over synthesized CNF formulas) (Si et al., 2018; Padhi & Millstein, 2017). We limit each method to one hour per problem in the same format as the SyGuS Competition (Alur et al., 2019). CLN2INV is able to solve all **124** problems in the benchmark. LoopInvGen solves **104** problems while Code2inv solves **90**.[3]

Figure 4a shows the measured runtime on each evaluated system. CLN2INV solves problems in **1.1** second on average, which is over **40×** faster than LoopInvGen, the second fastest system in the evaluation. It spends the most time on solver calls (0.6s avg.) and CLN training (0.5s avg.), with negligible time spent on preprocessing, data generation, and template generation on each problem ( less than 20ms ea.)[4].

In Table 1 we provide a more detailed analysis of how much time is spent on each stage of the invariant inference pipeline in CLN2INV. Measurements are averaged over 5 runs. The system spends most time on solver calls (0.6s avg.) and CLN training (0.5s on avg.) with negligible time spent on preprocessing, sampling, and template generation. For most problems in the Code2Inv benchmark, CLN2INV completes CLN training quickly (less than 0.2s) and spends most of its time performing solver checks, but it requires more epochs to train on some complex problems with many variables.

Table 1: Time spent on each stage of CLN2INV Pipeline.

|          | Preprocessing | Sampling | Template Gen. | CLN Training | Checking |
|----------|---------------|----------|---------------|--------------|----------|
| **Avg. Time** | 5ms       | 2ms      | 4ms           | 0.5s         | 0.6s     |
| **Max Time**  | 18ms      | 12ms     | 8ms           | 7.2s         | 5.3s     |

In general, CLN2INV has similar performance to LoopInvGen on simple problems but is able to scale efficiently to complex problems. Figure 4b shows the number of Z3 calls made by each method.

---

[3]The Code2Inv authors originally reported solving 92 problems using the same one hour timeout. We believe that the difference may be caused by changes in the testing environment or randomized model initialization.

[4]At most 6171 data points are generated for a given program in the benchmark, and 1041 data points per program on average, using the sampling strategy in §5.

Table 2: Results and summary statistics for performance evaluation.

| Method | Number Solved | Avg Time (s) | Avg Z3 Calls | Time/Z3 Call (s) |
|---|---|---|---|---|
| Code2Inv | 90 | 266.71 | 16.62 | 50.89 |
| LoopInvGen | 104 | 45.11 | 3,605.43 | 0.08 |
| CLN2INV | 124 | 1.07 | 31.77 | 0.17 |

For almost all problems, CLN2INV requires fewer Z3 calls than the other systems, although for some difficult problems it uses more Z3 calls than Code2Inv.

Table 2 summarizes results of the performance evaluation. Code2Inv require much more time on average per problem, but minimizes the number of calls made to an SMT solver. In contrast, Loop-InvGen is efficient at generating a large volume of guessed candidate invariants, but is much less accurate for each individual invariant. CLN2INV can be seen as balance between the two approaches: it searches over candidate invariants more quickly than Code2Inv, but generates more accurate invariants than LoopInvGen, resulting in lower overall runtime.

## 6.2 MORE DIFFICULT LOOP INVARIANTS

We consider two classes of more difficult loop invariant inference problems that are not present in the Code2Inv dataset. The first require conjunctions and disjunctions of multivariable constraints, and the second require polynomials with many higher order terms. Both of these classes of problems are significantly more challenging because they are more complex and cause the space of possible invariants to grow much more quickly.

To evaluate on problems that require invariants with conjunctions and disjunctions of multivariable constraints, we construct 12 additional problems. We specifically design these problems to contain loops with invariants that cannot be easily inferred with pre- and post-condition based heuristics. In this section Figure 5, we show one of these problems. By plotting the trace as shown in Figure 5b, it is easy to see that the points lie on one of the two lines expressible as linear equality constraints. The correct loop invariant for the program is $((t + u = 0) \vee (t - u = 0)) \wedge (u \leq 0)$.

```
//pre: t=-20/\u=-20
while (u != 0) {
  u++;
  if (t > 0)
    t = -t + 1;
  else
    t = -t - 1;
}
//post: t=0
```

(a) Pseudocode for Problem 1
(b) Plotted trace of program

Figure 5: Finding the loop invariant for Problem 1, which involves a disjunction of equalities

Another problem with multivariable conjunction is shown in Figure 6. In this program, we use `unknown` to denote an external function call that returns either true or false. As we cannot assume much about the function, we model the function call as sampling from a Bernoulli distribution with success probability $0.5$. Although the branching behavior is may not be deterministic, we know $(t + u = 0) \wedge (v + w = 0) \wedge (u + w \geq 0)$ is a correct invariant, as it holds regardless of which branch is taken. Our CLN2INV can learn this invariant within 20 seconds, while both Code2inv and LoopInvGen time out after one hour without finding a solution.

To evaluate on problems with higher order polynomial invariants, we test CLN2INV on the power summation problems in the form $u = \sum_{t=0}^{k} t^d$ for a given degree $d$, which have been used in evaluation for polyonomial loop invariant inference (Sharma et al., 2013; Nguyen et al., 2017). We discuss these problems in more detail in Appendix H. CLN2INV can correctly learn the invariant for 1st and 2nd order power summations, but cannot learn correct invariants for 3rd, 4th or 5th order

```
//pre: t=-10 /\ u=10 /\ v=-10 /\ w=10
while (u + w > 0) {
    if (unknown()) {
        t++; u--;
    } else {
        v++; w--;
    }
}
//post: t=w /\ u=v
```

Figure 6: Pseudocode for Problem 2, which involves a conjunction of equalities

summations, which have many more higher order terms. We do not evaluate the other methods on these problems because they are not configured for nonlinear arithmetic by default.

### 6.3 ABLATION STUDIES

**Effect of CLN Training on Performance.** CLN2INV relies on a combination of heuristics using static analysis and learning formulas from execution traces to correctly infer loop invariants. In this ablation we disable model training and limit CLN2INV to static models with no learnable parameters. Static CLN2INV solves 91 problems in the dataset. Figure 7 shows a comparison of full CLN2INV with one limited to static models. CLN2INV's performance with training disabled shows that a large number of problems in the dataset are relatively simple and can be inferred from basic heuristics. However, for more difficult problems, CLN learning is key to inferring correct invariants.

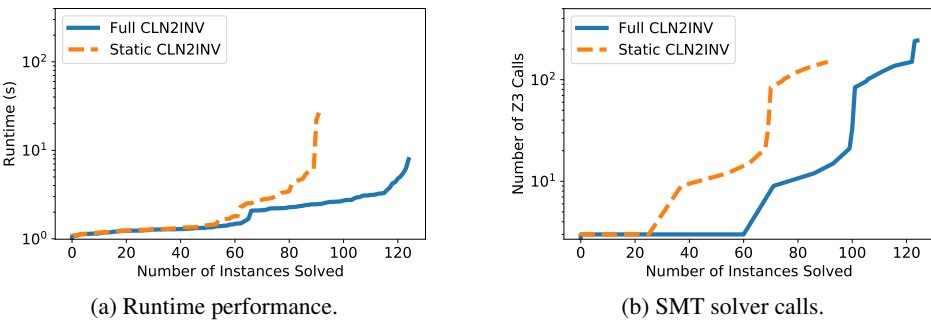

(a) Runtime performance.  (b) SMT solver calls.

Figure 7: Ablation study comparing static vs trained models.

## 7 CONCLUSION

We develop a novel neural architecture that explicitly and precisely learns SMT formulas by construction. We achieve this by introducing a new sound and complete semantic mapping for SMT that enables learning formulas through backpropagation. We use CLNs to implement a loop invariant inference system, CLN2INV, that is the first to solve all theoretically solvable problems in the Code2Inv benchmark and takes only 1.1 second on average. We believe that the CLN architecture will also be beneficial for other domains that require learning SMT formulas.

ACKNOWLEDGMENTS

This work is sponsored in part by NSF grants CNS-18-42456, CNS-18-01426, CNS-16-17670, CCF-1918400; ONR grant N00014-17-1-2010; an ARL Young Investigator (YIP) award; an NSF CAREER award; a Google Faculty Fellowship; a Capital One Research Grant; a J.P. Morgan Faculty Award; a Columbia-IBM Center Seed Grant Award; and a Qtum Foundation Research Gift. Any opinions, findings, conclusions, or recommendations expressed herein are those of the authors, and do not necessarily reflect those of the US Government, ONR, ARL, NSF, Google, Capital One J.P. Morgan, IBM, or Qtum.

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

## A  PROOF OF LIMIT OF $\mathcal{S}(t = u)$

$$\lim_{\substack{\epsilon \to 0^+ \\ B \cdot \epsilon \to +\infty}} \mathcal{S}(t = u) = \lim_{\substack{\epsilon \to 0^+ \\ B \cdot \epsilon \to +\infty}} \frac{1}{1 + e^{-B(t-u+\epsilon)}} \otimes \frac{1}{1 + e^{B(t-u-\epsilon)}} = \begin{cases} 1 & t = u \\ 0 & t \neq u \end{cases}$$

*Proof.* Let $f(t, u; B, \epsilon) = \frac{1}{1+e^{-B(t-u+\epsilon)}}$ and $g(t, u; B, \epsilon) = \frac{1}{1+e^{B(t-u-\epsilon)}}$. Then what we want to prove becomes

$$\lim_{\substack{\epsilon \to 0^+ \\ B \cdot \epsilon \to +\infty}} (f(t, u; B, \epsilon) \otimes g(t, u; B, \epsilon)) = \begin{cases} 1 & t = u \\ 0 & t \neq u \end{cases}$$

Because all $f, g, \otimes$ are continuous in their domain, we have

$$\lim_{\substack{\epsilon \to 0^+ \\ B \cdot \epsilon \to +\infty}} (f(t, u; B, \epsilon) \otimes g(t, u; B, \epsilon)) = \left( \lim_{\substack{\epsilon \to 0^+ \\ B \cdot \epsilon \to +\infty}} f(t, u; B, \epsilon) \right) \otimes \left( \lim_{\substack{\epsilon \to 0^+ \\ B \cdot \epsilon \to +\infty}} g(t, u; B, \epsilon) \right)$$

Using basic algebra, we get

$$\lim_{\substack{\epsilon \to 0^+ \\ B \cdot \epsilon \to +\infty}} f(t, u; B, \epsilon) = \begin{cases} 1 & t \geq u \\ 0 & t < u \end{cases} \qquad \lim_{\substack{\epsilon \to 0^+ \\ B \cdot \epsilon \to +\infty}} g(t, u; B, \epsilon) = \begin{cases} 1 & t \leq u \\ 0 & t > u \end{cases}$$

Combing these results, we have

$$\left( \lim_{\substack{\epsilon \to 0^+ \\ B \cdot \epsilon \to +\infty}} f(t, u; B, \epsilon) \right) \otimes \left( \lim_{\substack{\epsilon \to 0^+ \\ B \cdot \epsilon \to +\infty}} g(t, u; B, \epsilon) \right) = \begin{cases} 0 \otimes 1 & t < u \\ 1 \otimes 1 & t = u \\ 1 \otimes 0 & t > u \end{cases}$$

For any t-norm, we have $0 \otimes 1 = 0$, $1 \otimes 1 = 1$, and $1 \otimes 0 = 0$. Put it altogether, we have

$$\lim_{\substack{\epsilon \to 0^+ \\ B \cdot \epsilon \to +\infty}} (f(t, u; B, \epsilon) \otimes g(t, u; B, \epsilon)) = \begin{cases} 1 & t = u \\ 0 & t \neq u \end{cases}$$

which concludes the proof. □

## B  SOUNDNESS AND COMPLETENESS OF CLNS

In this section we provide formal descriptions and proofs of CLN soundness and completeness when trained to 0 loss. These properties are defined for CLNs as follows:

**Soundness.**  Given the SMT formula $F$, the CLN model $M_F$ constructed from $\mathcal{S}(F)$ always preserves the truth value of $F$. It indicates that given a valid assignment to the terms $\boldsymbol{x}$ in $F$, $F(\boldsymbol{x}) = True \iff M_F(\boldsymbol{x}) = 1$ and $F(\boldsymbol{x}) = False \iff M_F(\boldsymbol{x}) = 0$.

**Completeness.**  For any SMT formula $F$, a CLN model $M$ can be constructed representing that formula. In other words, CLNs can express all SMT formulas on integers and reals.

We formally state these properties in Theorem 1. Before that we need to define a property for t-norms.

**Property 1.** $\forall t\, u, (t > 0)$ and $(u > 0)$ implies $(t \otimes u > 0)$.

The product t-norm and Godel t-norm have this property, while the Lukasiewicz t-norm does not.
**Theorem 1.** For any quantifier-free linear SMT formula $F$, there exists CLN model $M$, such that

$$\forall \boldsymbol{x}\, B\, \epsilon,\ 0 \leq M(\boldsymbol{x}; B, \epsilon) \leq 1 \tag{1}$$

$$\forall \boldsymbol{x},\ F(\boldsymbol{x}) = True \iff \lim_{\substack{\epsilon \to 0^+ \\ B \cdot \epsilon \to \infty}} M(\boldsymbol{x}; B, \epsilon) = 1 \tag{2}$$

$$\forall \boldsymbol{x},\ F(\boldsymbol{x}) = False \iff \lim_{\substack{\epsilon \to 0^+ \\ B \cdot \epsilon \to \infty}} M(\boldsymbol{x}; B, \epsilon) = 0 \tag{3}$$

as long as the t-norm used in building $M$ satisfies Property 1.

*Proof.* For convenience of the proof, we first remove all $<, \leq, =$ and $\neq$ in $F$, by transforming $t < u$ into $\neg(t \geq u)$, $t \leq u$ into $\neg(t > u)$, $t = u$ into $(t \geq u) \wedge \neg(t > u)$, and $t \neq u$ into $(t > u) \vee \neg(t \geq u)$. Now the only operators that $F$ may contain are $>, \geq, \wedge, \vee, \neg$. We prove Theorem 1 by induction on the constructor of formula $F$. In the following proof, we construct model $M$ given $F$ and show that it satisfied Eq.(1)(2). We leave the proof for why $M$ also satisfied Eq.(3) to readers.

**Atomic Case.** When $F$ is an atomic clause, then $F$ will be in the form of $\boldsymbol{x} * W + b > 0$ or $\boldsymbol{x} * W + b \geq 0$. For the first case, we construct a linear layer with weight $W$ and bias $b$ followed by a sigmoid function scaled with factor $B$ and right-shifted with distance $\epsilon$. For the second case, we construct the same linear layer followed by a sigmoid function scaled with factor $B$ and left-shifted with distance $\epsilon$. Simply evaluating the limits for each we arrive at

$$\forall \boldsymbol{x}, F(\boldsymbol{x}) = True \iff \lim_{\substack{\epsilon \to 0^+ \\ B \cdot \epsilon \to \infty}} M(\boldsymbol{x}; B, \epsilon) = 1$$

And from the definition of sigmoid function we know $0 \leq M(\boldsymbol{x}; B, \epsilon) \leq 1$.

**Negation Case.** If $F = \neg F'$, from the induction hypothesis, $F'$ can be represented by models $M'$ satisfying Eq.(1)(2)(3). Let $p'$ output node of $M'$. We add a final output node $p = 1 - p'$. So $M(\boldsymbol{x}; B, \epsilon) = 1 - M'(\boldsymbol{x}; B, \epsilon)$. Using the induction hypothesis $0 \leq M'(\boldsymbol{x}; B, \epsilon) \leq 1$, we conclude Eq.(1) $0 \leq M(\boldsymbol{x}; B, \epsilon) \leq 1$.

Now we prove the " $\implies$ " side of Eq.(2). If $F(\boldsymbol{x}) = True$, then $F'(\boldsymbol{x}) = False$. From the induction hypothesis, we know $\lim_{\substack{\epsilon \to 0^+ \\ B \cdot \epsilon \to \infty}} M'(\boldsymbol{x}; B, \epsilon) = 0$. So

$$\lim_{\substack{\epsilon \to 0^+ \\ B \cdot \epsilon \to \infty}} M(\boldsymbol{x}; B, \epsilon) = \lim_{\substack{\epsilon \to 0^+ \\ B \cdot \epsilon \to \infty}} 1 - M'(\boldsymbol{x}; B, \epsilon) = 1 - 0 = 1$$

Next we prove the " $\impliedby$ " side. If $\lim_{\substack{\epsilon \to 0^+ \\ B \cdot \epsilon \to \infty}} M(\boldsymbol{x}; B, \epsilon) = 1$, we have

$$\lim_{\substack{\epsilon \to 0^+ \\ B \cdot \epsilon \to \infty}} M'(\boldsymbol{x}; B, \epsilon) = \lim_{\substack{\epsilon \to 0^+ \\ B \cdot \epsilon \to \infty}} 1 - M(\boldsymbol{x}; B, \epsilon) = 1 - 1 = 0$$

From the induction hypothesis we know that $F'(\boldsymbol{x}) = False$. So $F(\boldsymbol{x}) = \neg F'(\boldsymbol{x}) = True$.

**Conjunction Case.** If $F = F_1 \wedge F_2$, from the induction hypothesis, $F_1$ and $F_2$ can be represented by models $M_1$ and $M_2$, such that both $(F_1, M_1)$ and $(F_2, M_2)$ satisfy Eq.(1)(2)(3). Let $p_1$ and $p_2$ be the output nodes of $M_1$ and $M_2$. We add a final output node $p = p_1 \otimes p_2$. So $M(\boldsymbol{x}; B, \epsilon) = M_1(\boldsymbol{x}; B, \epsilon) \otimes M_2(\boldsymbol{x}; B, \epsilon)$. Since $(\otimes)$ is continuous and so are $M_1(\boldsymbol{x}; B, \epsilon)$ and $M_2(\boldsymbol{x}; B, \epsilon)$, we know their composition $M(\boldsymbol{x}; B, \epsilon)$ is also continuous. (Readers may wonder why $M1(\boldsymbol{x}; B, \epsilon)$ is continuous. Actually the continuity of $M(\boldsymbol{x}; B, \epsilon)$ should be proved inductively like this proof itself, and we omit it for brevity.) From the definition of $(\otimes)$, we have Eq.(1) $0 \leq M(\boldsymbol{x}; B, \epsilon) \leq 1$.

Now we prove the $\implies$ side of Eq.(2). For any $\boldsymbol{x}$, if $F(\boldsymbol{x}) = True$ which means both $F_1(\boldsymbol{x}) = True$ and $F_2(\boldsymbol{x}) = True$, from the induction hypothesis we know that $\lim_{\substack{\epsilon \to 0^+ \\ B \cdot \epsilon \to \infty}} M_1(\boldsymbol{x}; B, \epsilon) = 1$ and $\lim_{\substack{\epsilon \to 0^+ \\ B \cdot \epsilon \to \infty}} M_2(\boldsymbol{x}; B, \epsilon) = 1$. Then

$$\lim_{\substack{\epsilon \to 0^+ \\ B \cdot \epsilon \to \infty}} M(\boldsymbol{x}; B, \epsilon) = \lim_{\substack{\epsilon \to 0^+ \\ B \cdot \epsilon \to \infty}} M_1(\boldsymbol{x}; B, \epsilon) \otimes M_2(\boldsymbol{x}; B, \epsilon) = 1 \otimes 1 = 1$$

Then we prove the $\impliedby$ side. From the induction hypothesis we know that $M_1(\boldsymbol{x}; B, \epsilon) \leq 1$ and $M_2(\boldsymbol{x}; B, \epsilon) \leq 1$. From the non-decreasing property of t-norms (see §2.3), we have

$$M_1(\boldsymbol{x}; B, \epsilon) \otimes M_2(\boldsymbol{x}; B, \epsilon) \leq M_1(\boldsymbol{x}; B, \epsilon) \otimes 1$$

Then from the consistency property and the commutative property, we have

$$M_1(\boldsymbol{x}; B, \epsilon) \otimes 1 = M_1(\boldsymbol{x}; B, \epsilon)$$

Put them altogether we get

$$M(\boldsymbol{x}; B, \epsilon) \leq M_1(\boldsymbol{x}; B, \epsilon) \leq 1$$

Because we know $\lim_{\substack{\epsilon \to 0^+ \\ B \cdot \epsilon \to \infty}} M(\boldsymbol{x}; B, \epsilon) = 1$, according to the squeeze theorem in calculus, we get

$$\lim_{\substack{\epsilon \to 0^+ \\ B \cdot \epsilon \to \infty}} M_1(\boldsymbol{x}; B, \epsilon) = 1$$

From the induction hypothesis, we know that $F_1(\boldsymbol{x}) = True$. We can prove $F_2(\boldsymbol{x}) = True$ in the same manner. Finally we have $F(\boldsymbol{x}) = F_1(\boldsymbol{x}) \wedge F_2(\boldsymbol{x}) = True$.

**Disjunction Case.** For the case $F = F_1 \vee F_2$, we construct $M$ from $M_1$ and $M_2$ as we did in the conjunctive case. This time we let the final output node be $p = p_1 \oplus p_2$. From the continuity of $(\otimes)$ and the definition of $(\oplus)$ $(t \oplus u = 1 - (1 - t) \otimes (1 - u))$, $(\oplus)$ is also continuous. We conclude $M(\boldsymbol{x}; B, \epsilon)$ is also continuous and $0 \le M(\boldsymbol{x}; B, \epsilon) \le 1$ by the same argument as $F = F_1 \wedge F_2$.

Now we prove the " $\implies$ " side of Eq.(2). For any assignment $\boldsymbol{x}$, if $F(\boldsymbol{x}) = True$ which means $F_1(\boldsymbol{x}) = True$ or $F_2(\boldsymbol{x}) = True$. Without loss of generality, we assume $F_1(\boldsymbol{x}) = True$. From the induction hypothesis, we know $\lim_{\substack{\epsilon \to 0^+ \\ B \cdot \epsilon \to \infty}} M_1(\boldsymbol{x}; B, \epsilon) = 1$.

For any $(\oplus)$ and any $0 \le t, t' \le 1$, if $t \le t'$, then

$$t \oplus u = 1 - (1 - t) \otimes (1 - u) \quad \le \quad 1 - (1 - t') \otimes (1 - u) = t' \oplus u$$

Using this property and the induction hypothesis $M_2(\boldsymbol{x}; B, \epsilon) \ge 0$, we have

$$M_1(\boldsymbol{x}; B, \epsilon) \oplus 0 \quad \le \quad M_1(\boldsymbol{x}; B, \epsilon) \oplus M_2(\boldsymbol{x}; B, \epsilon) = M(\boldsymbol{x}; B, \epsilon)$$

From the induction hypothesis we also have $M_1(\boldsymbol{x}; B, \epsilon) \le 1$. Using the definition of $(\oplus)$ and the consistency of $(\otimes)$ $(0 \otimes x = 0)$, we get $M_1(\boldsymbol{x}; B, \epsilon) \oplus 0 = M_1(\boldsymbol{x}; B, \epsilon)$. Put them altogether we get

$$M_1(\boldsymbol{x}; B, \epsilon) \quad \le \quad M(\boldsymbol{x}; B, \epsilon) \le 1$$

Because we know $\lim_{\substack{\epsilon \to 0^+ \\ B \cdot \epsilon \to \infty}} M_1(\boldsymbol{x}; B, \epsilon) = 1$, according to the squeeze theorem in calculus, we get $\lim_{\substack{\epsilon \to 0^+ \\ B \cdot \epsilon \to \infty}} M(\boldsymbol{x}; B, \epsilon) = 1$.

Then we prove the " $\impliedby$ " side. Here we need to use the existence of limit:

$$\lim_{\substack{\epsilon \to 0^+ \\ B \cdot \epsilon \to \infty}} M(\boldsymbol{x}; B, \epsilon)$$

This property can be proved by induction like this proof itself, thus omitted for brevity.

Let

$$c_1 = \lim_{\substack{\epsilon \to 0^+ \\ B \cdot \epsilon \to \infty}} M_1(\boldsymbol{x}; B, \epsilon) \qquad\qquad c_2 = \lim_{\substack{\epsilon \to 0^+ \\ B \cdot \epsilon \to \infty}} M_2(\boldsymbol{x}; B, \epsilon)$$

Then

$$\lim_{\substack{\epsilon \to 0^+ \\ B \cdot \epsilon \to \infty}} M(\boldsymbol{x}; B, \epsilon) = \lim_{\substack{\epsilon \to 0^+ \\ B \cdot \epsilon \to \infty}} M_1(\boldsymbol{x}; B, \epsilon) \oplus M_2(\boldsymbol{x}; B, \epsilon) = c_1 \oplus c_2 = 1 - (1 - c_1) \otimes (1 - c_2)$$

Since we have $\lim_{\substack{\epsilon \to 0^+ \\ B \cdot \epsilon \to \infty}} M(\boldsymbol{x}; B, \epsilon) = 1$, we get

$$(1 - c_1) \otimes (1 - c_2) = 0$$

Using Property 1 of $(\otimes)$ (defined in §4), we have $c_1 = 1 \vee c_2 = 1$. Without loss of generality, we assume $c_1 = 1$. From the induction hypothesis, we know that $F_1(\boldsymbol{x}) = True$. Finally, $F(\boldsymbol{x}) = F_1(\boldsymbol{x}) \vee F_2(\boldsymbol{x}) = True$. $\qquad\square$

Careful readers may have found that if we use the continuous mapping function $\mathcal{S}$ in §3, then we have another perspective of the proof above, which can be viewed as two interwoven parts. The first part is that we proved the following lemma.

**Corollary 1.** *For any quantifier-free linear SMT formula F,*

$$\forall \boldsymbol{x} \; B \; \epsilon, \; 0 \leq \mathcal{S}(F; B, \epsilon)(\boldsymbol{x}) \leq 1$$

$$\forall \boldsymbol{x}, \; F(\boldsymbol{x}) = True \iff \lim_{\substack{\epsilon \to 0^+ \\ B \cdot \epsilon \to \infty}} \mathcal{S}(F; B, \epsilon)(\boldsymbol{x}) = 1$$

$$\forall \boldsymbol{x}, \; F(\boldsymbol{x}) = False \iff \lim_{\substack{\epsilon \to 0^+ \\ B \cdot \epsilon \to \infty}} \mathcal{S}(F; B, \epsilon)(\boldsymbol{x}) = 0$$

Corollary 1 indicates the soundness of $\mathcal{S}$. The second part is that we construct a CLN model given $\mathcal{S}(F)$. In other words, we translate $\mathcal{S}(F)$ into vertices in a computational graph composed of differentiable operations on continuous truth values.

## C  OPTIMALITY OF CLNS

**Optimality.** For a subset of SMT formulas (conjunctions of multiple linear equalities), CLNs are guaranteed to converge at the global minimum. We formally state this in Theorem 2. We first define another property similar to strict monotonicity.

**Property 2.** $\forall t_1 \; t_2 \; t_3, \; (t_1 < t_2)$ and $(t_3 > 0)$ implies $(t_1 \otimes t_3 < t_2 \otimes t_3)$.

**Theorem 2.** For any CLN model $M_F$ constructed from a formula, $F$, by the procedure shown in the proof of Theorem 1, if $F$ is the conjunction of multiple linear equalities then any local minimum of $M_F$ is the global minimum, as long as the t-norm used in building $M_F$ satisfies Property 2.

*Proof.* Since $F$ is the conjunction of linear equalities, it has the form

$$F = \bigwedge_{i=1}^{n} (\sum_{j=1}^{l_i} w_{ij} t_{ij} = 0)$$

Here $\mathbf{W} = \{w_{ij}\}$ are the learnable weights, and $\{t_{ij}\}$ are terms (variables). We omit the bias $b_i$ in the linear equalities, as the bias can always be transformed into a weight by adding a constant of 1 as a term. For convenience, we define $f(x) = \mathcal{S}(x = 0) = \frac{1}{1+e^{-B(x+\epsilon)}} \otimes \frac{1}{1+e^{B(x-\epsilon)}}$.

Given an assignment $\boldsymbol{x}$ of the terms $\{t_{ij}\}$, if we construct our CLN model $M_F$ following the procedure shown in the proof of Theorem 1, the output of the model will be

$$M(\boldsymbol{x}; \mathbf{W}, B, \epsilon) = \bigotimes_{i=1}^{n} f(\sum_{j=1}^{l_i} w_{ij} t_{ij})$$

When we train our CLN model, we have a collection of $m$ data points $\{t_{ij1}\}, \{t_{ij2}\}, ..., \{t_{ijm}\}$, which satisfy formula $F$. If $B$ and $\epsilon$ are fixed (unlearnable), then the loss function will be

$$L(\mathbf{W}) = \sum_{k=1}^{m} \mathcal{L}(M(\boldsymbol{x}; \mathbf{W}, B, \epsilon)) = \sum_{k=1}^{m} \mathcal{L}(\bigotimes_{i=1}^{n} f(\sum_{j=1}^{l_i} w_{ij} t_{ijk})) \tag{4}$$

Suppose $\mathbf{W}^* = \{w_{ij}^*\}$ is a local minima of $L(\mathbf{W})$. We need to prove $\mathbf{W}^*$ is also the global minima. To prove this, we use the definition of a local minima. That is,

$$\exists \delta > 0, \; \forall \mathbf{W}, \; ||\mathbf{W} - \mathbf{W}^*|| \leq \delta \implies L(\mathbf{W}) \geq L(\mathbf{W}^*) \tag{5}$$

For convenience, we denote $u_{ik} = \sum_{j=1}^{l_i} w_{ij} t_{ijk}$. Then we rewrite Eq.(4) as

$$L(\mathbf{W}) = \sum_{k=1}^{m} \mathcal{L}(\bigotimes_{i=1}^{n} f(u_{ik}))$$

If we can prove at $\mathbf{W}^*$, $\forall \; 1 \leq i \leq n, \; 1 \leq k \leq m, \; u_{ik} = 0$. Then because (i) $f$ reaches its global maximum at 0, (ii) the t-norm $(\otimes)$ is monotonically increasing, (iii) $\mathcal{L}$ is monotonically decreasing, we can conclude that $\mathbf{W}^*$ is the global minima.

Now we prove $\forall \, 1 \leq i \leq n, \, 1 \leq k \leq m, \, u_{ik} = 0$. Here we just show the case $i = 1$. The proof for $i > 1$ can be directly derived using the associativity of $(\otimes)$.

Let $\alpha_k = \bigotimes_{i=2}^{n} f(u_{ik})$. Since $f(x) > 0$ for all $x \in R$, using Property 2 of our t-norm $(\otimes)$, we know that $\alpha_k > 0$. Now the loss function becomes

$$L(\mathbf{W}) = \sum_{k=1}^{m} \mathcal{L}(f(u_{1k}) \otimes \alpha_k)$$

From Eq.(5), we have

$$\exists 0 < \delta' < 1, \, \forall \gamma, \, |\gamma| \leq \delta' \implies \sum_{k=1}^{m} \mathcal{L}(f(u_{ik}(1+\gamma)) \otimes \alpha_k) \geq \sum_{k=1}^{m} \mathcal{L}(f(u_{ik}) \otimes \alpha_k) \quad (6)$$

Because (i) $f(x)$ is an even function decreasing on $x > 0$ (which can be easily proved), (ii) $(\otimes)$ is monotonically increasing, (iii) $\mathcal{L}$ is monotonically decreasing, for $-\delta' < \gamma < 0$, we have

$$\sum_{k=1}^{m} \mathcal{L}(f(u_{ik}(1+\gamma)) \otimes \alpha_k) = \sum_{k=1}^{m} \mathcal{L}(f(|u_{ik}(1+\gamma)|) \otimes \alpha_k) \leq$$
$$\sum_{k=1}^{m} \mathcal{L}(f(|u_{ik}|) \otimes \alpha_k) = \sum_{k=1}^{m} \mathcal{L}(f(u_{ik}) \otimes \alpha_k) \quad (7)$$

Combing Eq.(6) and Eq.(7), we have

$$\sum_{k=1}^{m} \mathcal{L}(f(u_{ik}(1+\gamma)) \otimes \alpha_k) = \sum_{k=1}^{m} \mathcal{L}(f(u_{ik}) \otimes \alpha_k)$$

Now we look back on Eq.(7). Since (i) $\mathcal{L}$ is strictly decreasing, (ii) the t-norm we used here has Property 2 (see §4 for definition), (iii) $\alpha_k > 0$, the only case when (=) holds is that for all $1 \leq k \leq m$, we have $f(|u_{ik}(1+\gamma)|) = f(|u_{ik}|)$. Since $f(x)$ is strictly decreasing for $x \geq 0$, we have $|u_{ik}(1+\gamma)| = |u_{ik}|$. Finally because $-1 < -\delta' < \gamma < 0$, we have $u_{ik} = 0$. $\qquad\square$

# D  PRECONDITION STRENGTHENING

**Theorem 3.** Given a program $C$: assume($P$); while ($LC$) $\{C\}$ assert($Q$);
If we can find a loop invariant $I'$ for program $C'$: assume($P \wedge LC$); while ($LC$) $\{C\}$ assert($Q$);
and $P \wedge \neg LC \implies Q$, then $I' \vee (P \wedge \neg LC)$ is a correct loop invariant for program $C$.

*Proof.* Since $I'$ is a loop invariant of $C'$, we have

$$(P \wedge LC) \wedge LC \implies I' \quad (a) \qquad \{I' \wedge LC\}C\{I'\} \quad (b) \qquad I' \wedge \neg LC \implies Q \quad (c)$$

We want to prove $I' \vee (P \wedge \neg LC)$ is a valid loop invariant of C, which means

$$P \wedge LC \implies I' \vee (P \wedge \neg LC) \qquad \{(I' \vee (P \wedge \neg LC)) \wedge LC\}C\{I' \vee (P \wedge \neg LC)\}$$
$$(I' \vee (P \wedge \neg LC)) \wedge \neg LC \implies Q$$

We prove the three propositions separately. To prove $P \wedge LC \implies I' \vee (P \wedge \neg LC)$, we transform it into a stronger proposition $P \wedge LC \implies I'$, which directly comes from (a).

For $\{(I' \vee (P \wedge \neg LC)) \wedge LC\}C\{I' \vee (P \wedge \neg LC)\}$, after simplification it becomes $\{I' \wedge LC\}C\{I' \vee (P \wedge \neg LC)\}$, which is a direct corollary of (b).

For $(I' \vee (P \wedge \neg LC)) \wedge \neg LC \implies Q$, after simplification it will become two separate propositions, $I' \wedge \neg LC \implies Q$ and $P \wedge \neg LC \implies Q$. The former is exactly (c), and the latter is a known condition in the theorem. □

# E  IMPLEMENTATION DETAILS

**Training Data Generation Example.** Figure 8 provides an example of our training data generation procedure. The uninitialized variable $k$ is sampled according to the precondition $k \leq 8$ within the predefined width $r = 10$. So we end up enumerating $k = 8, 7, ..., -2$. For each $k$, the loop is executed repeatedly until termination, thus generating a small set of samples. The final training set is the union of these small sets.

```
//pre: t=10                        // run with k={-2..8}
//pre: u=0                         k = atoi(argv[1]);
//pre: k<=8                        t=10; u=0;
//invariant: 2t+u=20               while (t != k){
while (t != k){                        log(t, u, k);
   t = t - 1;                          t = t - 1;
   u = u + 2;                          u = u + 2;
}                                  }
//post: u=20-2k                    log(t, u, k);
```

    (a) The original loop program.                        (b) The sampling procedure.

Figure 8: Illustration of how training data is generated. After the sampling procedure in (b) we have a collection of 88 samples which will later be fed to the CLN model.

**Template Generation.** Templates are first generated from the pre- and post-conditions, followed by every pair of clauses extracted from the precondition, post-condition, and loop condition. Generic templates are then constructed consisting of one or more general equality constraints containing all variables conjoined with inequality constraints.

Three iterations of the generic template generation are shown here:

$$1 : (W_1 X = a_1) \wedge (x_1 \leq u_1) \wedge (x_1 \geq l_1) \wedge \cdots \wedge (x_n \leq u_n) \wedge (x_n \geq l_n)$$
$$2 : ((W_1 X = a_1) \wedge (W_2 X = a_2)) \wedge (x_1 \leq u_1) \wedge (x_1 \geq l_1) \wedge \cdots \wedge (x_n \leq u_n) \wedge (x_n \geq l_n)$$
$$3 : ((W_1 X = a_1) \vee (W_2 X = a_2)) \wedge (x_1 \leq u_1) \wedge (x_1 \geq l_1) \wedge \cdots \wedge (x_n \leq u_n) \wedge (x_n \geq l_n)$$

Algorithm 1 summarizes the template generation process. In it the following functions are defined:

| | |
|---|---|
| **construct_template:** | Construct a template given an smt formula. |
| **extract_clauses:** | Extract individual clauses from smt formulas. |
| **estimate_degrees:** | Performs log-log linear regression to estimate degree of each variable. |
| **polynomial_kernel:** | Executes polynomial kernel on variables and data for a given degree. |
| **is_single_constraint:** | Checks if condition is a single inequality constraint. |
| **extract_loop_constraint:** | Converts the loop condition to learnable smt template. |

Note that templates are generated on demand, so each template is used to infer a possible invariant before the next is generated.

---

**Algorithm 1 Template Generation Algorithm.**

---

**Input**:  $Pre \leftarrow$ precondition
$Post \leftarrow$ post-condition
$LC \leftarrow$ loop condition
$Max\_Template\_Len \leftarrow$ max template eq clauses
$vars \leftarrow$ vars in program
$X \leftarrow$ training data

1: construct_template($Post$)
2: construct_template($Pre$)
3: $clauses \leftarrow$ extract_clauses($Pre, Post, LC$)
4: **for** $(c_1, c_2)$ in all_pairs($clauses$) **do**
5:     construct_template($c1 \wedge c2$)
6:     construct_template($c1 \vee c2$)
7: **end for**
8: $max\_degree \leftarrow$ estimate_degrees($X$)
9: **if** $max\_degree > 1$ **then**
10:     $vars \leftarrow$ polynomial_kernel($vars, max\_degree$)
11: **end if**
12: $bound\_constraints \leftarrow ()$
13: **for** $var$ in $vars$ **do**
14:     $bound\_constraints \leftarrow bound\_contraints \wedge (var \leq u_{var})$
15:     $bound\_constraints \leftarrow bound\_contraints \wedge (var \geq l_{var})$
16: **end for**
17: **if** is_single_constraint($LC$) **then**
18:     $bound\_constraints \leftarrow bound\_constraints \wedge$ extract_loop_constraint($LC$)
19: **end if**
20: construct_template($((W \cdot vars = b)) \wedge bound\_constraints$)
21: $eq\_clauses \leftarrow [(W \cdot vars = b)]$
22: $template\_len \leftarrow 1$
23: **while** $template\_len \leq Max\_Template\_Len$ **do**
24:     **for** $eq\_clause$ in $eq\_clauses$ **do**
25:         construct_template($(eq\_clause \wedge (W \cdot vars = b)) \wedge bound\_constraints$)
26:         $eq\_clauses \leftarrow [eq\_clauses, eq\_clause \wedge (W \cdot vars = b)]$
27:         construct_template($(eq\_clause \vee (W \cdot vars = b)) \wedge bound\_constraints$)
28:         $eq\_clauses \leftarrow [eq\_clauses, eq\_clause \vee (W \cdot vars = b)]$
29:     **end for**
30:     $template\_len \leftarrow template\_len + 1$
31: **end while**

---

## F   PROPERTIES OF GAUSSIAN FUNCTION

We use a Gaussian-like function $S(t = u) = \exp(-\frac{(t-u)^2}{2\sigma_2})$ to represent equalities in our experiments. It has the following two properties. First, it preserves the original semantic of $=$ when $\sigma \to 0$, similar to the mapping $S(t = u) = \frac{1}{1+e^{-B(t-u+\epsilon)}} \otimes \frac{1}{1+e^{B(t-u-\epsilon)}}$ we defined in §3.

$$\lim_{\sigma \to 0^+} \exp(-\frac{(t-u)^2}{2\sigma^2}) = \begin{cases} 1 & t = u \\ 0 & t \neq u \end{cases}$$

Second, if we view $S(t = u)$ as a function over $t - u$, then it reaches its only local maximum at $t - u = 0$, which means the equality is satisfied.

## G   INVALID PROBLEMS FROM CODE2INV DATASET

Here we provide an example which is Problem 106 in the dataset. The post-condition $a \geq m$ is wrong if we start from $a = 0$, $m = 1$, and $k = 0$.

```
int k = 0;
int a, m;
assume(a <= m);
while (k < 1) {
    if (m < a) m = a;
    k = k + 1;
}
assert(a >= m);
```

Executing the program with these inputs results in $a = 0$, $m = 1$, and $k = 1$ as the if condition is never satisfied. But clearly, the post condition $a \geq m$ is violated. Below we tabulate the counterexamples invalidating the nine removed problems from the dataset:

Table 3: Invalid problems from Code2Inv dataset

| Problem | Counterexample starting state |
|---|---|
| 26 | $x, n = 0, 0$ |
| 27 | $x, n = 0, 0$ |
| 31 | $x, n = 0, 0$ |
| 32 | $x, n = 0, 0$ |
| 61 | $c, n = 0, 1$ |
| 62 | $c, n = 0, 1$ |
| 72 | $c, y = 0, 128$ |
| 75 | $c, y = 0, 128$ |
| 106 | $a, m, k = 0, 1, 0$ |

## H   POLYNOMIAL INVARIANTS

Here we provide results and an example of the higher order polynomial problems; more precisely, the power summation problems in the form $u = \sum_{t=0}^{k} t^d$ for a given degree $d$. We found that CLN2INV was able to learn invariants for programs which computes the sum of consecutive integers, which has 10 monomial terms and a maximum degree of 2. Since CLN2INV simply uses the polynomial kernel the number of terms quickly become unwieldy. We observe that CLN2INV struggles beginning with sum of squares, which has 20 monomial terms up to degree 3. Table 4 summarizes these results.

For further illustration, we describe in detail the case where $d$ is 3, i.e. sum of cubes.

Table 4: Results on power summation polynomial problems.

| Number Terms | Highest Degree | Solved? |
|:---:|:---:|:---:|
| 5 | 1 | ✓ |
| 10 | 2 | ✓ |
| 20 | 3 | ✗ |
| 35 | 4 | ✗ |
| 56 | 5 | ✗ |

```
//pre: t = u = 0 /\ k >= 0
while (t < k) {
  t++;
  u += t * t * t;
}
//post: 4u == k**2 * (k + 1)**2
```

Figure 9: Pseudocode for Polynomial Invariant Problem

The example loop in Figure 9 computes the sum of the first $k$ cubes. We know this sum has a closed form solution:

$$\sum_{i=0}^{k} k^3 = \frac{k^2(k+1)^2}{4} = \frac{k^4 + 2k^3 + k^2}{4}$$

For this problem, we would hope to extract the invariant:

$$(4u = t^4 + 2t^3 + t^2) \wedge (t <= k)$$

However, by naively using the polynomial kernel just as methods like NumInv suggest (Nguyen et al., 2017), we will have 35 monomials of degree at most four over three variables as candidate terms ($t^3 u, t^2 k^2, tu^2 k, ...$), and the model must learn to ignore all the terms except $u, t^4, t^3$, and $t^2$. We observe our model finds far more accurate coefficient for $t^4$ but than for lower ordered terms. We hypothesize that by nature of polynomials the highest order term is a good approximation for the whole polynomial. Thus, $u = t^4$ is a good approximation based on the data. The difficulty of learning polynomial invariants using CLN is an interesting direction for future studies.

