# OpenReview forum: "CLN2INV: Learning Loop Invariants with Continuous Logic Networks"
_ICLR.cc/2020/Conference — Accept (Poster)_

### Official Review · AnonReviewer2 · 2019-10-24
**Official Blind Review #2**

**Rating:** 8

**Review:**

The domain is loop invariant detection, in the static program analysis space. Loop invariants hold before, during, and after loop execution, and can be useful for compiler optimizations and/or correctness checking.
The paper explains Basic Fuzzy Logic and then uses it to introduce Continuous Satisfiability Modulo Theories, including proposed continuous mappings for inequalities (>, >=), negations, equalities, and requirements of t-norms to be useful for the relaxated optimization proposed. This Continuous Logic Network is then optimized to provide invariant proposals to Z3, an SMT solver. The whole system is used to solve the entire Code2Inv benchmark set, in substantially faster time and with fewer proposals to Z3 than comparable previous approaches. Ablations are provided which study the t-norm used (3 options considered), and another which uses heuristics only with no training/optimization to make static proposals to Z3.

On the whole, I like the presentation and the thinking here, and think it will be interesting to folks in the field, possibly spurring on further thinking in compilers, program synthesis, constrained optimization, etc, so recommend accepting.

Relaxed representations of satisfiability problems seems like something people have thought about in OR for some time, so I wonder if there is a missing part of the literature survey. A cursory glance turns up https://openreview.net/forum?id=BJxgz2R9t7

Interestingly, the heuristics do quite well, which calls into question how hard the dataset is, and how competitive the preceding works really were. Since these heuristics seem to be an important contributor to this approach, I think they deserve further discussion in the appendix, and/or source code should be released.

9 problems from the dataset are rejected as invalid. Please identify these in an appendix, and provide the counterexamples.

The dataset used here is quite small, and it seems like only ~30 of the problems are "hard" in requiring beyond-heuristic complexity. Couldn't the SyGuS tools be used to generate a much larger test set?

In fig 2, the model (x) doesn't match the template/invariant \/.

**Experience Assessment:**

I do not know much about this area.

**Review Assessment: Checking Correctness Of Derivations And Theory:**

I assessed the sensibility of the derivations and theory.

**Review Assessment: Checking Correctness Of Experiments:**

I assessed the sensibility of the experiments.

**Review Assessment: Thoroughness In Paper Reading:**

I read the paper at least twice and used my best judgement in assessing the paper.

---

> ### Author Response · Authors · 2019-11-06
> **Author Response to Review 2**
>
> We appreciate the thoughtful review and the detailed and constructive feedback!
>
> We agree that our work is related to OR work in relaxed SAT solving methods, and will extend our related work to address these approaches.
>
> We will specify the unsolvable problems and add counterexamples in the appendix as recommended.
>
> The dataset we use in our evaluation is from a state-of-the-art approach to learning loop invariants that was a NeurIPS 2018 spotlight (Si et al., 2018). Our current system was designed to operate on C programs, but we are working to extend it to work with problems that are described exclusively in an SMT formulation in order to operate on the Sygus competition benchmark, which contains 829 problems.
>
> We will also correct the error in Figure 2.

---

> ### Author Response · Authors · 2019-11-13
> **Updated Revision**
>
> Thank you for taking the time to review our submission and providing thoughtful suggestions. We have made revisions to the submission based on your earlier feedback as follows:
>
> 1.) As recommended, we have expanded out related works to specifically discuss relaxation efforts for satisfiability problems like Circuit-SAT.
>
> 2.) Additionally, we have tabulated counterexamples to the invalid problems in Appendix H.
>
> 3.) We have made our description of implementation more detailed in section 5, and added a detailed outline of our template generation algorithm (where the heuristics are incorporated) in Appendix F.
>
> 4.) We have fixed the error in Figure 2.

---

### Official Review · AnonReviewer4 · 2019-11-09
**Official Blind Review #4**

**Rating:** 3

**Review:**

Summary:

This paper introduces a novel way to find loop invariants for a
program. The loop invariants are expressed as SMT formulas. A
continuous relaxation of an SMT solver is introduces mapping
every SMT formula onto a truth value \in [0, 1]. This relaxation
called a continuous semantic mapping is decided such that every
true formula has a higher value than every false formula. This
allows an invariant to be learned.

Novelty and Significance:

This work is interesting and although authors do seem to be
outside of the community, I firmly believe is appropriate for
ICLR. If the claims made by the paper are true they constitute
a significant contribution to the field of program synthesis
and program analysis.

Technical Quality:

The evaluation was fairly thorough, but the paper can be
strengthened massively with a few small changes and additions.
It might be more helpful if there was a sense of how many problems
none of the systems can do and how complicated of a program can
this system extract a loop invariant from. Why were was it these
particular 12 that work? Are there examples that don't?

I don't know why all this time is spent on t-norms when behavior
on them is fairly similar and the simplest norm works best.

Lots of details are missing in this paper. How much training data
was generated? How long did it take? Does training data need to
be generated for each example? If so is that included in the
runtime for Figure 3?

The paper talks about neural architecture, but all I see is
effectively a curve-fitting task for some template. This feels
different from the code2inv paper where a program can be fed
into the system and the pretrained model emits the invariant.

Clarity:

Not enough of this paper concentrates on the novel aspects of the
approach. Section 5 discusses template generation, but not in
enough detail that I would be able to replicate this work. I
could also not find enough details in the Appendix.

I don't know why vital page space is spent on defining completeness
and soundness.


Possibly related work as relaxations to SAT/SMT solvers do exist in the literature.

Guiding High-Performance SAT Solvers with Unsat-Core Predictions
https://arxiv.org/abs/1903.04671

Learning to Solve SMT Formulas
https://papers.nips.cc/paper/8233-learning-to-solve-smt-formulas.pdf

Notes:

You cite Si et al. for LoopInvGen when you should be citing Pathi
and Millstein in the third paragraph on page 2.


**Experience Assessment:**

I have read many papers in this area.

**Review Assessment: Checking Correctness Of Derivations And Theory:**

I assessed the sensibility of the derivations and theory.

**Review Assessment: Checking Correctness Of Experiments:**

I carefully checked the experiments.

**Review Assessment: Thoroughness In Paper Reading:**

I read the paper at least twice and used my best judgement in assessing the paper.

---

> ### Author Response · Authors · 2019-11-10
> **Clarifying Questions**
>
> Thanks for taking the time to review our submission and providing thoughtful suggestions. We are working on a revision that incorporates all of your suggested edits, and will provide a detailed response with listing the changes when it is complete.
>
> We have two clarifying questions with regard to the requested edits:
>
> 1.) We want to double check your concern about the discussion of t-norms and soundness/completeness taking too much space is with regard to the paper body (and not the proofs in the appendix). If so, we can certainly consolidate those sections and move more details to the appendix.
>
> 2.) In order to provide sufficient details about the implementation and experiments we are considering putting full descriptions in the appendix (though we will fit as much as possible into the paper body). Will this be ok?

---

> > ### Comment · AnonReviewer4 · 2019-11-10
> > **Response to Clarification**
> >
> > I'm bringing up the t-norms since I want more detail in Section 5 and since I want your paper to stay within the page limits I'm suggesting some of the content on t-norms can be moved into the Appendix or honestly simply removed.
> >
> > I would appreciate if as much of the details can be in the paper body but use your best judgement about what goes where.
> >
> > Also to help save time, I am deeply concerned that if a new dataset needs to be created for each benchmark problem that runtime should really reflect that. Otherwise it's a bit deceptive to say solving the problem only takes a second. I may be misunderstanding this so I'm repeating this is a major concern of mine.

---

> > > ### Author Response · Authors · 2019-11-10
> > > **Response to Concern about Data Generation Time**
> > >
> > > Thank you for your quick response! Just a quick clarification about your main concern.
> > > The reported numbers *already include* the data generation time. The data generation (sampling) procedure for the C programs in the code2inv dataset is very fast—it takes on average 1.9 milliseconds and generates 1041 samples. The longest sampling time for a benchmark program is 11.0 milliseconds to generate 6171 samples. Also, just to make sure that we are on the same page, note that our system requires no pre-training and infers the loop invariant for each program directly based on the corresponding sampled data.
> > >
> > > We will provide full details of our sampling procedure and a breakdown of the time spent on each stage of the pipeline in the updated version.

---

> > > ### Author Response · Authors · 2019-11-13
> > > **Updated Revision**
> > >
> > > Thank you for taking the time to review our submission and providing thoughtful suggestions. We have uploaded a revised version based on your feedback with the following updates:
> > >
> > > `1.) To address the limitations of our approach, we have added a discussion in Section 6.2 Paragraph 3 on page 8, and Appendix J with examples that CLN2INV cannot currently solve.
> > >
> > > 2.) As suggested, we simplified the t-norm section and moved more of our formal discussion on soundness and completeness from the body to the appendix in order to allocate more space for describing our implementation and experiments.
> > >
> > > 3.) We have added details about the amount of time our system spends on each stage of its pipeline on the Code2Inv benchmark in Section 6.1 Paragraph 2 on page 7 with additional details in Appendix I, and also added details about the number of samples generated in Section 6.1 as a footnote on page 7.  We additionally added an example showing how training data generation is performed in Appendix F.
> > >
> > > 4.) We have clarified why we consider CLNs to be a general purpose neural architecture in Section 4 Paragraph heading “CLN Construction” on page 5.
> > >
> > > 5.) In Section 5 Paragraphs 1-4, pages 5-6, we added a detailed description of the preprocessing, training data generation, and the template generation process as requested to make the experiment reproducible. To ensure absolute clarity on our procedure, we go into further details on the template generation algorithm in Appendix F.
> > >
> > > 6.) We have updated the related work on page 2 to address neural network relaxation techniques used previously for SAT/SMT, including fastSMT and the work on NeuroSAT for unsat-core detection by Selsam and Bjørner.
> > >
> > > 7.) We have corrected the LoopInvGen citation on page 2.

---

### Public Comment · ~Charles_Sutton1 · 2020-05-20
**Tautologies and negative training examples?**

Thanks for this interesting paper. I have a question about the learning objective in Section 4.

In the training data, all traces will satisfy the invariant. Therefore, it seems that the learning method will like to generate tautologies; more formally if there exists weights W s.t. for *all* x, M(x; W, B, epsilon) = 1, this will be globally optimal for the learning objective. Is this correct?

If so, have the authors observed such solutions being produced in practice? It is possible that many of the SMT templates generated by Code2Inv do not have W that would make them tautologies, but the learning method could still "overgenerate" as it were, causing the end-to-end method to miss potential invariants.

---

> ### Author Response · Authors · 2020-05-21
> **Addressing Tautologies in Practice**
>
> This is an excellent point. It is possible for the model to learn tautologies such as $(0*x_1 + 0*x_2 + \cdots + 0*1 = 0)$ depending on the structure of the template. However, we found in practice that given random nonzero weight initializations, the model never converged to these degenerate solutions on loops in the evaluation.
>
> In our more recent work in PLDI 2020, Learning Nonlinear Loop Invariants with Gated Continuous Logic Networks (https://arxiv.org/pdf/2003.07959.pdf), we provide a more principled solution to learning complex invariants while avoiding degenerate cases through weight normalization and gating.

---

### Decision · Program_Chairs · 2019-12-19

**Decision:**

Accept (Poster)

**Comment:**

This paper implements a novel architecture for inferring loop invariants in verification (though the paper bridges to compilers).  The idea is novel and the paper is well executed.  It is not the usual topic for ICLR, but not presents an important application of deep learning done well, and it has interesting implications for program synthesis.  Therefore, I recommend acceptance.